# Mastocytosis and Skin Cancer: The Current State of Knowledge

**DOI:** 10.3390/ijms24129840

**Published:** 2023-06-07

**Authors:** Agnieszka Kaszuba, Martyna Sławińska, Jakub Żółkiewicz, Michał Sobjanek, Roman J. Nowicki, Magdalena Lange

**Affiliations:** Department of Dermatology, Venereology and Allergology, Medical University of Gdańsk, Smoluchowskiego Street 17, 80-214 Gdańsk, Poland

**Keywords:** mastocytosis, melanoma, non-melanoma skin cancer, pathogenesis, diagnostics, treatment

## Abstract

Mastocytosis is a heterogeneous group of diseases associated with excessive proliferation and accumulation of mast cells in different organs. Recent studies have demonstrated that patients suffering from mastocytosis face an increased risk of melanoma and non-melanoma skin cancer. The cause of this has not yet been clearly identified. In the literature, the potential influence of several factors has been suggested, including genetic background, the role of cytokines produced by mast cells, iatrogenic and hormonal factors. The article summarizes the current state of knowledge regarding the epidemiology, pathogenesis, diagnosis, and management of skin neoplasia in mastocytosis patients.

## 1. Introduction

Mastocytosis is a heterogeneous group of diseases associated with excessive proliferation and accumulation of mast cells (MSc) in different organs. The most commonly affected organs are bone marrow, the skin, the liver, the spleen, and the lymph nodes [1]. Recent studies have demonstrated that patients suffering from mastocytosis face an increased risk of melanoma and non-melanoma skin cancer (NMSC) [2,3,4]. The cause of this has not yet been clearly identified.

In the literature, the potential influence of several factors has been suggested, including genetic background [5,6,7,8,9], the role of cytokines produced by MCs [10,11,12,13,14,15], iatrogenic [16], and hormonal factors [17,18].

The aim of this study was to summarize the current state of knowledge regarding the pathogenesis, diagnosis, and management of melanoma and NMSC in patients with mastocytosis.

## 2. Mastocytosis—Classification and Epidemiology

*Urticaria pigmentosa*, (UP) currently termed maculopapular cutaneous mastocytosis, was first described in 1869 by Nettleship and Tay [19]. In 1936, Sezary first described the first case of mastocytosis [20,21].

In the decades since, the classification of the disease, diagnostic criteria, and treatment approach have evolved.

According to the current World Health Organization (WHO) classification published in 2016, mastocytosis is divided into cutaneous mastocytosis (CM), systemic mastocytosis (SM), and mast cell sarcoma [22].

Mastocytosis is a rare disorder, and there is no accurate data on the frequency of the disease. The estimated prevalence of mastocytosis (both systemic and cutaneous) is 1:10,000. The prevalence of systemic mastocytosis in Europe is 1 in every 8000 to 10,000 [23]. In a population-based study conducted in the United States, the incidence of systemic mastocytosis in adults was higher among Caucasians compared to African Americans (0.056 vs. 0.018 per 100,000) [24].

Mastocytosis is a disease that can occur at any age. In adults, the systemic form is predominant, whereas in children, it is usually limited to the skin, with a tendency to resolve spontaneously around puberty [25,26].

Most patients with mastocytosis experience only cutaneous involvement, and maculopapular cutaneous mastocytosis (*urticaria pigmentosa*) is the most commonly diagnosed variant of the disease, occurring in about 80% of patients [25,27,28].

## 3. Epidemiological Link between Mastocytosis and Skin Cancer

Numerous publications discuss comorbidities in patients with mastocytosis, including solid cancers (especially melanoma and NMSC) and cardiovascular diseases (mostly venous thromboembolism [VTE] [2,3,4]).

Until January 2023, nine case reports [3,4,16,29,30,31,32,33,34,35] and two population studies [2,3] analyzing correlations between the incidence of mastocytosis, melanoma, and NMSC have been published in the PubMed database. The summary of these findings is presented in Table 1.

Olsen et al. [2] conducted a population-based cohort study that included 687 Danish patients (older than 14 years) diagnosed with systemic mastocytosis (with or without *urticaria pigmentosa*) and 68,700 controls from the general population. In this registry-based study, the incidence of solid cancers, VTE, myocardial infarction (MI), and stroke was analyzed during a 15-year period (1997–2012). It was found that the melanoma risk was about 7.5 times higher and the NMSC risk was around 2.5 times higher in mastocytosis patients compared to the general population. Additionally, melanoma was more advanced at the time of diagnosis compared to the controls (stage I-II diagnosis in 64.3% of SM patients vs. 79.3% in controls; stage III–IV in 14.3% vs. 12.0%, respectively; unknown stage in 21.4% vs. 8.8%, respectively). As discussed by the authors, SM patients are frequently managed by dermatologists, which could influence the detection of skin malignancy. It should be emphasized that the study described only the incidence of melanoma and NMSC in patients with mastocytosis without multivariate analysis including other known melanoma risk factors, such as ultraviolet radiation (UV) treatments, and concerned an exclusively Caucasian population. Interestingly, the study did not show an increased risk of breast cancer, lung cancer, or colorectal cancer [2].

In another study, Hagglund et al. [3] examined a group of 81 Swedish patients diagnosed with SM between 2007 and 2011. Melanoma was diagnosed in four patients (three of whom were also diagnosed with UP). The study showed that in patients diagnosed with SM, there was a 5% risk of developing melanoma, compared to 1.2–1.6% in the general Swedish population. The researchers emphasized that none of the patients had been treated with phototherapy in the past.

The remaining data come from case reports/case series and record 14 melanoma cases in 13 patients (4 males and 9 females), with a mean Breslow thickness of 1.13 mm (median 0.55 mm [0.1–5.6 mm]). In this group, most patients were diagnosed with cutaneous mastocytosis and six with SM. Only one of the reported patients received psolaren ultraviolet radiation (PUVA) therapy [3,4,16,29,30,31,32,33,34].

## 4. Pathogenetic Link between Mastocytosis and Skin Cancer

The precise pathogenetic link explaining the increased risk of skin cancer in mastocytosis patients has not been elucidated. In the literature, the potential contribution of several factors has been suggested, including genetic background, cytokines, neuropeptides, and hormonal and iatrogenic factors.

### 4.1. Genetic Factors

#### KIT

MCs and melanocytes originate from hematopoietic stem cells and neural crest cells, respectively. Both cell lines express KIT [36,37,38]. Dysregulation of KIT occurs in multiple diseases. Besides mastocytosis and melanoma, it is also observed in gastrointestinal stromal tumors (GISTs) [39], lung cancer, acute myeloid leukemia, and germ cell tumors [40].

*KIT* encodes the Kit receptor, which is physiologically activated by its ligand—Stem Cell Factor (SCF). SCF is produced in a variety of cells, including fibroblasts and endothelial cells. Interactions between the KIT receptor and SCF are responsible for the recruitment of MC progenitors into tissues, the regulation of proliferation and survival, and the activation of mastocytes [1,41]. Somatic mutations that occur in *KIT* in the course of mastocytosis cause constitutive KIT activation even in the absence of SCF, which finally leads to the clonal proliferation of MCs in different organs. The most common *KIT* mutation, present in 80–90% of patients with SM, is an aspartic acid to valine substitution (D816V *KIT* mutation) [8,9,14].

As shown in the previous studies, mutations of *KIT* also occur in melanoma, though less frequently. According to the study by Pham et al., 3% of all melanoma cases harbor *KIT* mutations [42]. Interestingly, they are relatively frequent in melanoma of special anatomical locations—being reported in 30–39% of cases of mucosal melanoma, 20–36% of acral melanoma, and in 20–28% of melanoma cases developing on chronically sun-damaged skin [43,44,45]. In melanoma, *KIT* mutations show heterogeneous distribution through the gene, and they are detected most frequently in exon 11 (L576P) and exon 13 (K642E) [46].

Notably, the D816V *KIT* mutation has rarely been reported in melanoma [47]. In the AACR project, it was detected in 4 out of 785 melanoma cases [48].

It has been shown that Imatinib may be effective in patients with melanoma harboring c-KIT alterations [49]. Independently, a recently published study on the animal model showed that tumor-infiltrating MCs are associated with resistance to anti-PD-1 therapy, and combining anti-PD-1 with sunitinib/imatinib results in depletion of mast cells, leading to complete regression of melanoma [50].

Other studies showed that *KIT* mutations can also be indirectly associated with melanocyte proliferation. A mutated KIT receptor forms a protein complex with microphthalmia-associated transcription factor (MITF) and SRC family kinases, leading to excessive activity [5]. Activated MITF affects the overexpression of genes related to melanocyte proliferation and survival (such as *TBX2*, *BCL2*, *SOX10*, *CDK2*, *HIF1A*, *P35*, and *DIAPH1*) [51].

Excessive activity of SRC kinases results in abnormal activation of nuclear transcription factors—signal transducer and activator transcription 3 (STAT3) and signal transducer and activator transcription 5 (STAT5)—and ultimately leads to increased proliferation of MCs and melanocytes [11].

### 4.2. Tumor Microenvironment

It has been suggested that MCs may promote the development of cancer by releasing mediators conducive to tumor development, angiogenesis, tissue remodeling, and affecting the adaptive immune response [10,52,53].

Among mediators produced by MCs, three groups have been distinguished, including pre-formed substances (serotonin, histamine, heparin, tryptase, and chymase), molecules synthesized after mastocyte stimulation (PAF, PDG2, and LTB4 and LTD4), and cytokines (IL-1, IL-3, IL-5, IL-8, IL-10, GM-CSF, TNF-α, TGF-β, and VEGF) [41].

Previous studies showed that MCs are present in the tumor microenvironment, including in basal cell carcinoma (BCC), squamous cell carcinoma (SCC), and melanoma [53,54,55]. Patients diagnosed with BCC and melanoma have been found to have a higher density of dermal MCs [15,56,57]. Other studies showed a correlation between oral SCC progression and the increasing number of MCs within the tumor tissue [58].

The interaction between inflammatory and neoplastic cells in melanoma and NMSCs is complex [59,60,61,62]. It is interesting whether the presence of MCs in mastocytosis patients is secondary to the presence of skin malignancy or whether the increased number of MCs in the skin of patients with mastocytosis may lead to a higher risk of oncogenesis.

Interestingly, a study by Hart et al. [63] showed that MCs may contribute to the development of BCC through their immunosuppressive effect on sun-exposed skin. On the other hand, it has been shown that neoplastic cells can produce chemotactic factors for MCs, including IL-8 [15] and RANTES [64], and, as a result, induce recruitment of MCs into the tumor microenvironment. It was also noticed that melanoma cells, through the cytokines secreted into their microenvironment (TGF-β and IL-1) and their influence on upregulating C3 expression, induce a change in the MC phenotypes. This leads to increased secretion of cytokines, which promote tumor progression [15].

The interaction between immune and cancer cells in tumor stroma is complex but has been shown to be a crucial element affecting tumor progression in melanoma and NMSC. MCs present in the tumor microenvironment can exert both pro- and anti-tumor properties [65].

The immunosuppressive activity of MCs is connected with IL-10, TNF-α, and histamine [52]. On the other hand, mastocytes release cytokines with anti-tumor activity (IL-1, IL-4, and IL-6).

There is growing data connecting the expression of PD-1 receptors on MCs with tumor progression. Inhibitors of the programmed cell death protein 1 (PD-1) radically improved survival times in patients with metastatic melanoma. Nevertheless, a significant proportion of patients are resistant to PD-1 inhibition (for anti-PD-1 monotherapy with nivolumab/pembrolizumab it is up to 60–70%, and for combination therapy with anti-CTLA-4 ipilimumab, it is 40–50%). Little is known about the predictive factors of this resistance [66].

Recently, Li et al. [67] showed that MCs in the tumor microenvironment may be a poor prognostic factor connected with resistance to anti-PD-1 immunotherapy. According to the study, PD-1 antibodies activate MCs and lead to the release of histamine and tumor-promoting cytokines, which may subsequently reduce the effect of immunotherapy. Additionally, a stabilizer of the MC membrane, cromolyn sodium, was found to inhibit this effect. The authors suggested that inhibiting MC degranulation could be an effective solution to PD-1 immunotherapy resistance.

Other studies have also documented the role of histamine and tryptase in mastocyte-related tumor progression. It has been found that histamine released by MCs can induce tumor cell proliferation through H1 receptors. In contrast, suppression of the immune system is connected with interaction with H2 receptors [68].

This association is interesting in the context of a study by Fritz et al. [69], who found an association between improved survival in melanoma patients and treatment with desloratadine (HR = 0.46; 95% CI 0.29–0.73, *p* = 0.001) or loratadine (HR = 0.50; 95% CI 0.28–0.88, *p* = 0.02). Additional observations concerned the reduced risk of secondary cutaneous melanoma in patients receiving one of the aforementioned drugs.

Interestingly, tryptase exhibits pro-angiogenic activity through its ability to degrade the connective tissue matrix by activating matrix metalloproteinases (MMPs) [70] and activating PAR-2 receptors located on endothelial cells [71,72].

MCs may also stimulate neoangiogenesis in melanoma and NMSC via the production of VEGF [64] and IL-6 [73]. Moreover, other MC products (heparin, histamine, tryptase, chymase, TGF-β, FGF-2, and IL-8) may stimulate endothelial cell proliferation [11,13].

### 4.3. Hormones and Neuropeptides as a Potential Link between Mastocytosis and Skin Cancer

#### 4.3.1. Sex Hormones

Mastocytosis may affect both women and men, being slightly more frequent in males among children and more prevalent in females after puberty. Although the role of sex hormones remains unclear in mastocytosis pathogenesis, they are suspected to affect the course of the disease and possibly impact the risk of developing skin cancer. There are many discrepancies between the results of studies evaluating the influence of pregnancy on the symptoms of mastocytosis. Though the majority of papers describe deterioration or stabilization of the clinical course of mastocytosis [74,75,76], there was one study reporting clinical improvement and cessation of symptoms during pregnancy [77].

Evidence from in vitro studies shows activation of MCs by sex hormones. MCs express both estrogen and progesterone receptors on their surfaces. Estradiol leads to increased synthesis and release of MC mediators in vitro [77,78]. A study performed by Kirmaz et al. [79] proved that menstrual cycle-dependent alteration of sex hormone serum concentrations impacts the results of skin prick tests (displaying highest reactivity to allergens mid-cycle), most probably due to estrogen-related augmentation of MCs’ degranulation processes. The role of progesterone is less well documented. Progesterone seems to counteract estrogens, inhibiting MCs’ degranulation [80] and implying that the relative ratio of estrogen to progesterone receptors on MCs’ surfaces may be responsible for an uncertain disease prognosis during pregnancy. Interestingly, despite the expression of androgen receptors in MCs, testosterone treatment does not lead to MC degranulation [81].

Tamoxifen, a selective estrogen receptor modulator, was shown to inhibit MCs’ degradation and proliferation of human NMSC [82]. The effect of tamoxifen on melanoma evaluated in clinical trials is equivocal [83]. In patients with indolent systemic mastocytosis, tamoxifen showed limited efficacy [84].

There is no data regarding the effect of hormonal replacement therapy (HRT) on mastocytosis. However, HRT was shown to significantly increase the risk of melanoma (OR = 1.21) [85], BCC (HR = 1.16) [86], and SCC (IRR = 1.35 with every 5 years of using HRT) [87].

In recent years, the number of reports indicating the relationship between sex hormones and the development of melanoma and NMSC has grown [88]. Data from the nation-wide Norway Cancer Registry show that melanoma is the most common neoplasm during pregnancy and lactation [89]. In one epidemiological study on skin tumors among Dutch patients with mastocytosis, in a cohort of 269 mastocytosis patients in which 8 developed melanoma, males were significantly more frequently affected than females when compared to the non-melanoma group (*p* = 0.03) [90]. In another study analyzing 81 patients with systemic mastocytosis, 4 cases of melanoma were reported without sex predilection. However, both females diagnosed with melanoma were of premenopausal age (under 45 years old), whereas both males were over 60 years old [3]. This is in accordance with the European Cancer Information System, which estimated a higher incidence of melanoma among females younger than 45 years of age and a nearly two-fold greater melanoma risk in males older than 75 years of age [91]. Hitherto, studies have led to the conclusion that sex hormones may make females prior to menopause more susceptible to melanoma and imply possible involvement of sex hormones in melanoma, NMSC pathogenesis, and the clinical course of mastocytosis. Unfortunately, data regarding the sex of patients with melanoma and NMSC were not provided in one of the largest epidemiological studies on systemic mastocytosis [2]. Conversely, the summary of extracted cases (Table 1) shows a higher incidence of melanoma in females with mastocytosis. Hence, the low number of cases and available demographic data do not allow for unequivocal conclusions on the sex-related risk of melanoma in mastocytosis patients.

#### 4.3.2. Hypothalamus-Pituitary-Adrenal (HPA) Axis

In a study by Antoniewicz et al. [92], significantly elevated serum concentrations of adrenocorticotropic hormone (ACTH) were reported in patients with mastocytosis when compared to the control group. These findings support the idea of an altered skin HPA (sHPA) axis in patients with mastocytosis. Moreover, Antoniewicz et al. underlined the potential role of α-Melanocyte Stimulating Hormone (α-MSH), supposedly responsible for skin lesion pigmentation in CM (*urticaria pigmentosa*). Along with ACTH, α-MSH arises from the proteolytic cleavage of proopiomelanocortin peptide, which is synthesized in the pituitary gland. Furthermore, both ACTH and α-MSH have been identified in MC granules. However, the role of α-MSH in the pathogenesis of melanoma is unclear. It was shown that α-MSH may act as an anti-inflammatory agent. On the other hand, it may concurrently prevent the recognition of melanoma by the immune system [93]. When it comes to NMSC, a study by Slominski et al. [94] showed that keratinocytes may be stimulated by ACTH and MSH and, as a consequence, facilitate the development of BCC.

Together with ACTH and MSH, corticotropin-releasing hormone (CRH) is another hormone produced by the pituitary gland that has also been localized inside MC secretory granules. Furthermore, high serum CRH concentration and CRH receptor presence were identified on MCs in a specimen obtained from a patient who underwent acute psychological stress with a subsequent exacerbation of CM (*urticaria pigmentosa*). Additionally, MCs were reported to secrete CRH and express CRH receptors, indicating the role of CRH in stress-related symptoms of mastocytosis [95]. Activation of CRH receptors on MCs may induce angiogenesis, thus facilitating the progression of the disease [96]. CRH receptors have been identified in melanoma cell lines but not in benign melanocytic nevi [97]. In a study performed by Yang et al. [98], CRH was found to be involved in the migration of melanoma cells in an animal model. Moreover, CRH was reported to inhibit human keratinocyte proliferation [99], thus pointing to it possibly not being involved in NMSC development.

#### 4.3.3. Hypothalamus-Pituitary-Thyroid (HPT) Axis

Among an abundance of different molecules, mast cells can also synthesize and store the thyroid hormones triiodothyronine (T3) and thyroid stimulating hormone (TSH) [100]. Homeostasis of thyroid hormones (THs) is conditioned in an autocrine fashion by the presence of deiodinases, which have the ability to activate and deactivate thyroid hormones. Deiodinases are dynamically expressed in a tissue-specific manner and are essential in skin development and maturation [101]. The impact of deiodinases on skin cancer has been most extensively studied in an animal model of BCC. For instance, type 3 deiodinase (THs inactivating enzyme) is overexpressed in BCC, thus resulting in local hypothyroidism leading to increased keratinocyte proliferation. Moreover, it was shown that type 3 deiodinase-knocked-down mice display a 5-fold reduction in the growth of BCC xenografts [102]. The effect of histamine on mast cell T3 concentration seems peculiar. Mast cell exposition to an attomolar (10^−18^) (but not higher) concentration of histamine results in elevated T3 content in mast cells, implying the existence of a hormonal network in the immune system [103]. Furthermore, TSH increases the mast cell concentration of T3 [104]. Despite T3 reducing SCC cell proliferation, it concurrently induces epithelial–mesenchymal transition, enhancing SCC invasiveness [105].

Apparently, the impact of THs also occurs via genomic action through the TSH receptor (TSHR). It has been shown that skin cells, namely keratinocytes, fibroblasts, and epidermal melanocytes, express the TSH receptor (TSHR). Ellerhorst et al. [106] demonstrated that the expression of TSHR is higher in malignant and premalignant melanocytic lesions and displayed that melanoma cells, but not normal melanocytes, are induced to proliferate at a physiologically relevant concentration of TSH. A study by Shah et al. [107] exhibited a higher prevalence of hypothyroidism among melanoma patients compared to the general population. Contrary to this, an animal model of hypothyroid-induced mice with uveal melanoma displayed significantly longer survival times compared to mice receiving thyroxine [108]. The link between mastocytosis, skin cancer, and the HPT axis seems to be even more evident considering that mast cells express and store TSH [100]. However, the complex and often incoherent data allow only the assumption that a reduced concentration of THs is protective, whereas an increased TSH level may promote the proliferation of melanoma cells.

#### 4.3.4. Neuropeptides

Substance P (SP) is a neuroendocrine peptide that binds to the neurokinin 1 receptor (NK1-R). Binding of SP to NK1-R results in the activation of several signaling pathways that promote cell proliferation, angiogenesis, and apoptosis inhibition, which enhance neoplasm growth and metastasis. Skin lesions in patients with mastocytosis have significantly upregulated expression of NK1-R compared to healthy controls [109]. Interestingly, the presence of NK1-R has also been described in melanoma, dysplastic, and Spitz nevi cells, but not in acquired benign melanocytic nevi [110]. The serum concentration of SP is elevated in patients with mastocytosis compared to healthy controls [109]. Consequently, the promotion of malignant transformation of melanocytes and tumor progression may be facilitated in subjects with mastocytosis. It was also shown that the NK1-R antagonist is effective in inducing melanoma cell apoptosis in vitro, and therefore it may be a candidate for the treatment of melanoma [110]. Moreover, SP elicits pruritus, thus implying NK1-R antagonists as a promising therapy for itch in mastocytosis [111].

Another neuropeptide that has the ability to activate and is simultaneously found in MC granules is calcitonin gene-related peptide (CGRP). Although CGRP is mainly recognized for its potent vasodilatory properties, it also serves as an immunomodulatory agent in the skin and a regulator of keratinocyte growth [112]. CGRP attenuates Langerhans cell antigen presentation in the animal model, which may be the culprit for impaired anti-tumoral immune response [113]. Interestingly, intradermal injection of a CGRP antagonist was proven to avert this immunosuppressive effect of CGRP [114]. Additionally, CGRP was demonstrated to stimulate the proliferation of melanocytes and promote melanin production, which could potentially play a role in the observed basal hyperpigmentation in individuals with CM [115]. The impact of CGRP on melanoma is uncertain; CGRP was shown to inhibit melanogenesis and induce apoptosis in vitro [116]. However, CGRP augments the exhaustion of cytotoxic CD8+ T cells, which in turn curtails their ability to eliminate melanoma [117]. CGRP was shown to promote oral SCC and other NMSC development and plays a key role in the augmentation of pain related to neoplasm progression [118]. Moreover, the potential involvement of CGRP in skin cancer in mastocytosis patients is even more apparent considering that its concentration is elevated in the sera of patients with mastocytosis [109].

Additionally, the role of sensory nerve endings has to be noted, as they may bind the relationship between mastocytosis and skin neoplasms. Sensory nerve stimulation (e.g., by injury or UVR) results in the secretion of CGRP and SP, which in turn results in MC activation and degranulation [119]. Moreover, nerve endings may directly enhance the release of histamine from the adjacent MCs. Together with SP, CGRP may act synergistically in releasing histamine, which has been shown to be involved in skin cancer development [120].

It transpires that ultraviolet radiation, an independent risk factor for skin cancer, may not only be responsible for MC activation [121], but may also act as a mediator in the crosstalk between mast cells and dermal cells via β-endorphin, another neuropeptide that may activate and be retrogradely secreted by MCs. In the animal model, UV exposure triggered the release of β-endorphin from keratinocytes, consequently acting as an indirect addictive agent for sun-seeking behaviors [122]. Given that MCs may release β-endorphin and be further activated by keratinocyte-derived β-endorphin after UV exposure, they may contribute to tanning dependence, thus enhancing the self-perpetuating process of UV-related increased risk of developing melanoma and NMSC.

### 4.4. Mastocytosis Treatment and the Risk of Skin Malignancy

According to one hypothesis, some therapeutic modalities used in the treatment of mastocytosis may influence the risk of malignancy. The main goal of mastocytosis treatment is to control the secretion of mediators or their effects [123]. Second-generation H1 antihistamines are crucial drugs in the anti-mediator therapy of all forms of mastocytosis [124].

It has been reported that drugs such as desloratadine and loratadine have been associated with improved melanoma survival and decreased risk of secondary melanoma in melanoma patients [69].

In contrast, both PUVA (psoralen ultraviolet A radiation) and UVB 311 (phototherapy with 311 nm wavelength ultraviolet B radiation), which can be used in mastocytosis to relieve skin symptoms like itching [34], are known to be risk factors for melanoma and NMSC [125,126,127].

Table 2 summarizes the treatment methods that are used in mastocytosis and their influence on the risk of NMSC and melanoma.

## 5. Screening and Management of Skin Cancer and Melanoma in Mastocytosis Patients

Despite the above-discussed increased risk of melanoma and NMSC in patients with mastocytosis, none of the current guidelines address the need for routine screening in this group of patients [1,25,28].

In daily practice, such screening includes mainly visual inspection and dermoscopic assessment. It has been shown that dermoscopy increases diagnostic sensitivity and specificity in melanoma diagnosis and allows for more precise detection of early melanoma compared to unaided eye examination [141].

To date, there have been no published studies on the dermoscopic aspect of melanocytic nevi in mastocytosis patients. It is important to underline that maculopapular mastocytosis lesions and melanocytic nevi may both present with brown reticular lines (pigment network) [142] (Figure 1).

Patients who have a collision of melanocytic/non-melanocytic lesions with cutaneous mastocytosis lesions maybe more difficult to assess dermoscopically (Figure 2 and Figure 3).

Finally, patients may misinterpret signs of cutaneous malignancy as mastocytosis skin involvement (Figure 4).

As the basis of skin cancer treatment and melanoma is surgical excision, clinicians managing patients with mastocytosis should be familiar with perioperative risk in these patients in order to avoid complications, as it is known that patients with mastocytosis have a higher risk of perioperative anaphylaxis [143,144].

## 6. Discussion

Recent studies demonstrate that patients suffering from mastocytosis face an increased risk of melanoma and NMSC. The cause of this phenomenon has yet to be clearly identified.

The precise pathogenetic link explaining the increased risk of skin cancer in mastocytosis patients has not been elucidated. In the literature, the potential contribution of several factors has been suggested, including genetic background, cytokines, hormones, neuropeptides, and iatrogenic factors.

Though the role of *KIT* alternations has been established in mastocytosis and melanoma, mutations show heterogeneous distribution through the gene in both diseases. It is possible that *KIT* mutations can be indirectly associated with melanocyte proliferation (e.g., via MITF and SRC family kinases). Additionally, recent studies have shown an association between tumor-infiltrating MCs and resistance to anti-PD-1 therapy in melanoma.

It has been suggested that MCs may promote the development of cancer by releasing mediators conducive to tumor development, angiogenesis, tissue remodeling, and affecting the adaptive immune response. The interactions between MCs and neoplastic cells in the tumor microenvironment are complex. It has been shown that MCs may contribute to NMSC through their immunosuppressive effect on sun-exposed skin. On the other hand, neoplastic cells may produce cytokines that recruit MCs or induce changes in their phenotype. The immunosuppressive activity of MCs is connected, i.e., with IL-10, TNF-α, and histamine.

Sex hormones may make females prior to menopause more susceptible to melanoma and possibly imply a clinical course of mastocytosis. Unfortunately, large-scale epidemiological data regarding the sex of patients with melanoma and NMSC are unavailable.

Our understanding of the link between the hypothalamus-pituitary-adrenal axis and neuropeptides (substance P, calcitonin gene-related peptide, and β-endorphin) in skin cancer and mastocytosis is poor, and the matter requires further study. The roles of the hypothalamus-pituitary-thyroid axis, mastocytosis, and skin neoplasms are complex and trilateral.

Among the treatment modalities used in mastocytosis patients are methods that may increase the risk of skin malignancy (i.e., systemic glucocorticosteroids, phototherapy), but which also possibly increase survival in melanoma (i.e., disodium cromoglycate, desloratadine, loratadine, and fexofenadine).

Data from population studies supports the need for active screening for skin neoplasia in patients with mastocytosis. In this context, cooperation between dermatologists, allergologists, and hematologists is crucial—regular medical visits associated with mastocytosis management should create an opportunity for total body skin examination and patient education concerning self-examination and the rules of photoprotection.

## 7. Conclusions

The existing literature data concerning the pathogenesis, diagnosis, management, and prognosis of melanoma and NMSC in patients with mastocytosis are scarce, and this topic requires further studies.

Data from population studies supports the need for active screening for skin neoplasia in patients with mastocytosis. In this context, cooperation between dermatologists, allergologists, and hematologists seems crucial—regular medical visits associated with mastocytosis management should create an opportunity for total body skin examination and patients’ education concerning self-examination and the rules of photoprotection. Possible treatment with phototherapy should be planned carefully, balancing the potential benefits and risks of this form of treatment.

## Figures and Tables

**Figure 1 ijms-24-09840-f001:**
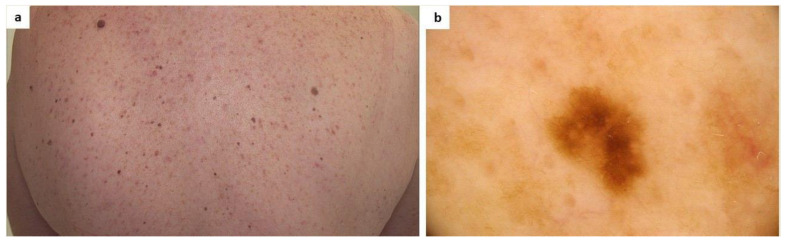
(**a**) A patient diagnosed with maculopapular cutaneous mastocytosis (MCM) and multiple melanocytic nevi. Clinically, it may be difficult to differentiate some nevi from mastocytosis skin lesions; (**b**) Dermoscopy shows a melanocytic nevus surrounded by areas of a light-brown pigmented network typical of MCM (FotoFinder, Medicam 800 HD, ×20).

**Figure 2 ijms-24-09840-f002:**
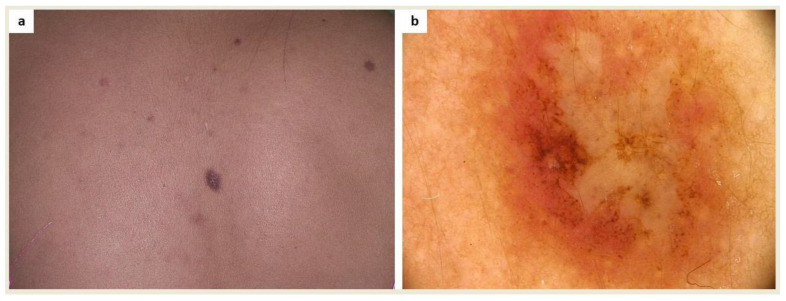
(**a**) A patient diagnosed with maculopapular cutaneous mastocytosis (MCM) and multiple melanocytic nevi was referred for surgical excision due to an atypical skin lesion in the interscapular area; (**b**) Dermoscopy revealed asymmetry of color and structures as well as a polymorphic vascular pattern (FotoFinder, Medicam 800 HD, ×20). Based on histopathological evaluation, a collision of melanocytic compound nevus and mastocytoma was diagnosed.

**Figure 3 ijms-24-09840-f003:**
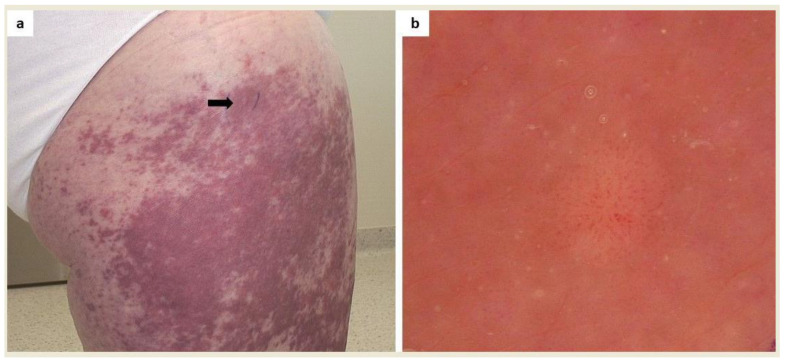
(**a**) A patient diagnosed with systemic mastocytosis with cutaneous involvement who had noticed an amelanotic nodule within coalescing mastocytosis skin lesions on the right thigh (arrow); (**b**) Dermoscopy showed a polymorphic vascular pattern (a non-specific sign of malignancy) which led to diagnostic excision (FotoFinder, Medicam 800 HD, ×20). Histopathological examination showed a cumulation of mastocytes.

**Figure 4 ijms-24-09840-f004:**
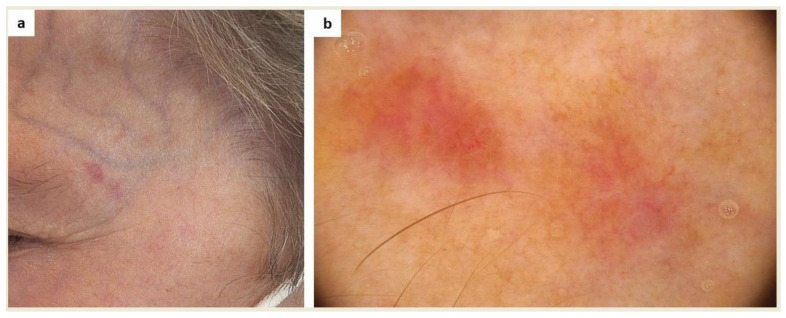
(**a**) This amelanotic lesion in the left temporal area was noticed in a patient with systemic mastocytosis with cutaneous involvement during a routine dermoscopic examination. The patient considered it a sign of cutaneous mastocytosis. (**b**) Dermoscopy showed a polymorphic vascular pattern (a non-specific sign of malignancy), erosion, and white-pink and light-brown structureless areas (FotoFinder, Medicam 800 HD, ×20). Based on histopathological examination, a diagnosis of basal cell carcinoma was made.

**Table 1 ijms-24-09840-t001:** Summary of case reports describing patients with mastocytosis and malignant melanocytic tumors.

First Author, Journal, Year of Publication	Patient’s Age/Gender	Ethnicity	Mastocytosis Variant	Tryptase Concentration [ng/mL]	Diagnosis/Breslow Depth [mm]	Tumor Location	Mastocytes in the Same Histopathological Sample	History of Phototherapy	Additional Information
Todd, Clin Exp Dermatol, 1991, [4]	71/F	Caucasian	UP (since 35 yo) SM (since 55 yo)	NR	Malignant melanoma (Ulcerated, vertical growth phase predominant with superficial spreading element); 5.6	right axilla	peripheral mastocytes	no	no family history of melanoma; no childhood sunburns; no dysplastic nevi; died 2 years after diagnosis of melanoma
Wallenfang, Der Hautarzt, 2001, [16]	39/F	NR	UP	NR	Nodular melanoma; 1.9	right thigh		yes	
Kowalzic, J Dtsch Dermatol Ges, 2009, [34]	62/F	NR	TMEP	6.04	Superficial spreading melanoma (in pre existing nevus); 0.65	left forearm	mastocytes infiltrating melanoma	no	obesity
Donati, J Dermatol Case Rep, 2014 [29]	38/F	Caucasian	UP	NR	superficial spreading melanoma; 0.5	left arm	mastocytes with perivascular infiltration in the papillary and reticular dermis	no	eruptive melanocytic nevi
Hagglund, Acta Derm Venereol, 2014, [3]	40/F	NR	SM + UP	70.0	Metastatic melanoma in lymph node, unknown primary origin; -	NR		no	
	45/M	NR	SM + UP	36.0	Nodular melanoma; 1.4	NR		no	
	20/F	NR	SM + UP	62.0	Malignant melanoma: NR	NR		no	
	63/F	NR	SM	190.0	Malignant melanoma not otherwise specified, 0.4	NR		no	
Ruini, Der Hautarzt, 2018, [30]	49/M	NR	UP/SM? Unknown (due to bone marrow involvement of melanoma)	16.4	Superficial spreading melanoma; 0.55 (primary or skin metastasis)	left groin		no	Died 10 weeks after diagnosis of melanoma
Capo, Clin Exp Dermatol, 2018, [31]	62/M	Caucasian	UP	NR	Superficial spreading melanoma; 0.45	NR		no	Simultaneous diagnosis UP and melanoma
Akdogan, J Cosmet Dermatol, 2020, [32]	51/M	NR	UP	35.6	Superficial spreading melanoma; 0.7	right arm		no	
Rydberg, Int J Dermatol, 2020, [33]	50/F	Caucasian	TMEP	29.4	Metastatic melanoma; unknown primary origin	right arm		no	previous history of melanoma
	54/F	Caucasian	TMEP	11.0	Superficial spreading melanoma; 0.1	right arm		no	Melanoma diagnosis in patient’s father
					Superficial spreading melanoma: 0.2	right ankle			

F—female, M—male, UP—urticaria pigmentosa, NR—no response, NM—nodular mastocytosis.

**Table 2 ijms-24-09840-t002:** Data on treatment methods used in mastocytosis and the risk of melanoma and non-melanoma skin cancer.

Mastocytosis Treatment Method	Cutaneous Melanoma Risk	Basal Cell Carcinoma Risk	Cutaneous Squamous Cell Carcinoma Risk
H1-antihistamines	desloratadine, loratadine, and fexofenadine improve survival in cutaneous malignant melanoma [69]	NR	NR
H2-antihistamines	clemastine and possibly cetirizine could potentially worsen survival in cutaneous malignant melanoma [69]		
antileukotrienes	NR	NR	NR
disodium cromoglycate	induces melanoma cells necrosis [128] improves the efficacy of PD-1 immunotherapy in vivo [67]	NR	NR
inhibitors of the proton pump	reduce the survival of patients treated with ipilimumab and nivolumab but not with ipilimumab alone [129]	NR	NR
systemic glucocorticosteroids	no influence [130]	increased risk [131,132]	increased risk [131,132]
phototherapy (UVA1, narrow-band UVB, PUVA)	increased risk [125,126]	increased risk [133]	increased risk [134]
omalizumab	NR	NR	NR
cytoreductive therapy with cladribine (2-chlorodeoxyadenosine—2-CdA)	there is no significant risk of malignant melanoma [135]	NR	NR
interferon-α	adjuvant therapy in patients with melanoma at high risk of recurrence after surgical resection [136]	BCC treatment option	SCC treatment option
imatinib	melanoma treatment with KIT amplifications and/or mutations (no response in KIT D816V mutation) [137]	NR	NR
masitinib	no effectiveness in prospective clinical trial [138]	NR	NR
midostaurin	midostaurin did not show clinical or biologic activity against metastatic melanoma [139]	NR	NR
miltefosine	Miltefosine caused a specific topical regression of the cutaneous metastases of the aggressive melanoma in one patient [140]	NR	NR

NR—not reported.

## Data Availability

Not applicable.

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
