# Peer review of "Mastocytosis and Skin Cancer: The Current State of Knowledge"

_ijms, 2023, doi:10.3390/ijms24129840_

Round 1

Reviewer 1 Report

The abstract

The abstract is well structured and emphasizes the study’s main idea while stirring an interest in its target reading audience . It also explains the rationale that led to this study and  it conveys the study utility in future medical practice. Furthermore, there is a decidely academic use of english language and the authors present us with a fluent text. 

Main body of work 

The article is structured into six chapters,  each of them being signaled accordingly and arranged in a logical order:

Introduction Mastocytosis – classification and epidemiology Epidemiological link between mastocytosis and skin cancer Pathogenetic link between mastocytosis and skin cancer Screening and management of skin cancer and melanoma in mastocytosis patient Conclusions

The introduction is coherent, well-composed and documented, and it sets the context the data that has been reviewed

The key papers discussed are appropriately cited and most of the latter have been published in the last 10 to 15 years. The figures, being pictures or tables are properly described and annotated.  Another strength of the study consists of it signaling the fields in which a proper assessment has been impossible. 

The conclusion are clear in formulation and concise, while being fully supported by the data previously presented. They also enhance how important further studies in this field could be in providing a new understanding and management of patients with mastocytosis. 

Quality of English language

The manuscript should not be language edited prior to acceptance.

Overview 

In conclusion, this article consists of valuable research which might change the clinical management of patients with mastocytosis and may stimulate further research in the field. 

Overall, the manuscript should be accepted without further editing.

Author Response

The authors would like to thank to the reviewer for the positive feedback on the manuscript.

Reviewer 2 Report

The authors of ”Mastocytosis and skin cancer − the current state of knowledge” aimed  to summarize the current state of knowledge regarding the pathogenesis, diagnosis and management of melanoma and NMSC in patients with mastocytosis.

They presented data regarding  mastocytosis – classification and epidemiology, epidemiological link between mastocytosis and skin cancer, pathogenetic link between mastocytosis and skin cancer, screening and management of skin cancer and melanoma in mastocytosis patient.

The four figures sustains the importance of this pathology.

.

Author Response

(The authors gave the same response as above.)

Reviewer 3 Report

The manuscript entitled Mastocytosis and skin cancer − the current state of knowledge” by Agnieszka Kaszuba et al., was presented as an interesting Review belonging to the “Molecular Oncology” section, Special Issue “Skin Cancer and Melanoma”. The manuscript provides a good overview of the actual knowledge regarding the pathogenesis, diagnosis and management of melanoma and non-melanoma skin cancer in patients with mastocytosis. The manuscript is well-written. Some paragraphs lack bibliographic citations. More concise language would make the article easier to read. Although the Review was presented in a comprehensive and interesting way, it lacks a more detailed conclusions section. Please, improve the conclusion, perhaps adding a discussion section. Moreover, it would be interesting to further develop the section “4.3. Hormones and neuropeptides as a potential link between mastocytosis and skin cancer”. It should be better stressed the role of hormones in the pathogenesis of mastocytosis and skin cancer. With regard to the role of hormonal influence in the increased risk of melanoma and non-melanoma skin cancer in patient suffering from mastocytosis, the authors take into consideration that recent studies demonstrated that not only sex hormones but also thyroid hormones signaling (not only via type 2 and 3 deiodinases, but also via genomic action through thyroid hormone receptors) influence the propensity to develop skin pathologies, in particular basal and squamous cell carcinoma, both belonging to non-melanoma skin cancer (PMID: 33509032; PMID: 31913269; PMID: 32111151). Moreover, abundant literature demonstrated that Mast Cells express thyroid hormone receptors and Thyrotropic hormone (TSH), and can store thyroid hormones, suggesting that thyroid function affects Mast Cells function (PMID: 30983971; PMID: 19205847; PMID: 19706378; PMID: 30150993). Consider the possibility to extend the review also to the Hypothalamus-Pituitary-Thyroid (HPT) axis.

Based on personal proficiency in English, the reviewer retains that only a Minor editing of the English language is required.

Author Response

The authors would like to thank to the reviewer for the positive feedback on the manuscript.

According to the reviewer’s suggestion the section “4.3. Hormones and neuropeptides as a potential link between mastocytosis and skin cancer” has been updated with information from the recommended articles.

According to the reviewer’s suggestion that the manuscript lacks a more detailed conclusions section we have improved the conclusion section and added disscusion.

Lacking bibliographic citations have been added to the appropriate paragraphs.

Due to minor lingual mistakes indicated to be corrected by the reviewer, the article underwent language-editing performed by native speaker.

Round 2

Reviewer 3 Report

The authors have addressed most of my concerns. The reviewer approves the manuscript for the publication in IJMS in the current state.